# New Kind of Polymer Materials Based on Selected Complexing Star-Shaped Polyethers

**DOI:** 10.3390/polym11101554

**Published:** 2019-09-24

**Authors:** Andrzej Szymon Swinarew, Beata Swinarew, Jadwiga Gabor, Magdalena Popczyk, Klaudia Kubik, Arkadiusz Stanula, Zbigniew Waśkiewicz, Thomas Rosemann, Beat Knechtle

**Affiliations:** 1Institute of Materials Science, Faculty of Computer Science and Materials Science, University of Silesia, 75 Pułku Piechoty 1A, 41-500 Chorzów, Poland; andrzej.swinarew@us.edu.pl (A.S.S.); jadwiga.gabor@us.edu.pl (J.G.); magdalena.popczyk@us.edu.pl (M.P.); klaudia.k.kubik@gmail.com (K.K.); 2Institute for Engineering of Polymer Materials & Dyes, Paint & Plastics Department; Chorzowska 50a, 44-100 Gliwice, Poland; b.swinarew@impib.pl; 3Institute of Sport Science, Department of Exercise and Sport Performance, The Jerzy Kukuczka Academy of Physical Education, 40-065 Katowice, Poland; a.stanula@awf.katowice.pl; 4Institute of Sport Science, Department of Team Sport Games, The Jerzy Kukuczka Academy of Physical Education, 40-065 Katowice, Poland; z.waskiewicz@awf.katowice.pl; 5Department of Sports Medicine and Medical Rehabilitation, Sechenov University, Moscow 119991, Russia; 6Institute of Primary Care, University of Zurich, 8091 Zurich, Switzerland; thomas.rosemann@usz.ch; 7Medbase St. Gallen Am Vadianplatz, 9001 St. Gallen, Switzerland

**Keywords:** complexing polyethers, drug-sensitive materials, acetylsalicylic acid, star-shaped polyethers

## Abstract

In today’s analytical trends, there is an ever-increasing importance of polymeric materials for low molecular weight compounds including amines and drugs because they can act as carriers or capture amines or drugs. The use of this type of materials will allow the development of modern materials for the chromatographic column beds and the substrates of selective sensors. Moreover, these kinds of materials could be used as a drug carrier. Therefore, the aim of this study is presenting the synthesis and complexing properties of star-shaped oxiranes as a new sensor for the selective complexation of low molecular weight compounds. Propylene oxide and selected oxirane monomers with carbazolyl in the substituent were selected as the monomers in this case and tetrahydrofuran as its solvent. The obtained polymer structures were characterized using the MALDI-TOF. It was found that in the initiation step potassium hydride deprotonates the monomer molecule and takes also part in the nucleophilic substitution. The resulting polymeric material preferably cross-linked with selected di-oxiranes (1,2,7,8-diepoksyoktan in respect ratio 3:1 according to active center) was then used as a stationary phase in the column and thin layer chromatography for amine separation and identification. Sorption ability of the resulting deposits was determined using a quartz microbalance (QCMB). The study was carried out in stationary mode and flow cells to simulate actual operating phase conditions. Based on changes in electrode vibration frequency, the maximum amount of adsorbed analyte and the best conditions for its sorption were determined.

## 1. Introduction

A very large group of compounds used in the study are macrocyclic compounds, among others synthetic macrocyclic polyethers. Synthetic macrocyclic polyethers have been known since 1937 when Lüttringhaus [1] prepared oxo-crown ethers from 1,3-dihydroxybenzene. In 1967 Pedersen [2] found that macrocyclic polyethers had a high affinity for alkali- and alkaline earth metal ions. That discovery involved the development of coordination chemistry of metal ions, which are strongly and selectively complexed by selected cryptands or cryptands’ groups. The aim of research in this area is the selection of compounds for the stationary phases in columns for UHPLC or TLC plates, the controlled transport of ions or as materials for applications in oncology. The possibility of design compounds for use in the construction of analytical tools for sport fair play control and molecular devices requires understanding the relations between their structure and properties.

Complexing compounds are very popular because of the wide application range. They are used among other things: in medicine as a capture and neutralize metal ions compounds [3,4,5], as catalysts in polymerization reactions [6,7,8,9,10,11,12,13], as crown ether complex cation ionic liquids catalysts [14,15,16], as activators [17,18], as protecting groups in organic synthesis [19], as chiral gels utilized as memory-erasing recycle system or logic gates based on the stimuli-responsive gel–sol transition, as chiral selectors to separate amino acid enantiomers in capillary electrophoresis analysis of amino acid enantiomers [20,21,22,23], as effective sensors [24,25,26,27,28,29], as a component of ion-selective membrane electrode [30,31], as extractants for various elements [32,33,34,35,36], and as hydraphiles—synthetic ionophores designed to mimic some properties of protein channels that conduct cations such as sodium [37].

There has been a strong demand for the resolution of chiral lipophilic amines in the pharmaceutical area. These receptors also allowed to reveal the mechanisms of influence amines in biological systems [38]. Complexing agents have a great future, and further research in this area will lead to significant improvements in the development of science.

The first analyzed compounds capable of binding metal ions and ammonium salts were crown ethers. Further studies have been carried out on combining ethers with various types of substituents; they have led to the creation of optically active molecules that allow separation of enantiomers. One proved that the type of substituent (ring, open chain) affects the flexibility of the molecule. The use of substituent in a ring, which is also a chiral barrier, increases the rigidity of the chiral cavity in the crown ether and thereby enhances its ability to detect of enantiomers. The phenolic group containing inner-ring hydroxyl group forms in the ether cavity able to combine neutral amines. Moreover, adding an extra electron type substituent in the para position changes the acidity of the phenyl group [39]. Naemura et al. studying this effect have concluded that the ether complexation ability of neutral amines and their stability strongly depend on the acidity, but if in the ether occur strongly acceptor or donor groups, then there is the possibility of capturing of a wide range of neutral amines [39]. The aim of this study was to determine complexation ability of acetylsalicylic acid, ethylenediamine, and aniline by the polymers obtained, which in the future could contribute to their use in the previously mentioned applications. These amines were chosen because they represent their class in a very good way and their presence was possible to confirm by means of possessed analytical equipment. The analysis was made using a quartz microbalance fixed and flow, using the 5 MHz crystals and 10 MHz. Because synthesized materials are not from the group of conducting polymers to the quartz applied drops of the polymer in a solvent and after evaporation of the solvent, the layer was analyzed. In the analyses, the following solvents were used: tetrahydrofurane (THF), acetonitrile, phosphate buffer, pH 7.6, demineralized water.

## 2. Materials and Methods

Studies of the reaction glycidol (96%; Fluka, CAS Number: 556-52-5) with potassium hydride (30 wt % dispersion in mineral oil; Sigma-Aldrich, CAS Number: 7693-26-7) was carried out in the THF (99,8%; Acros organic, CAS Number: 109-99-9) solution containing ether 18-crown-6 (18C6) (Merck, CAS Number: 17455-13-9) with equimolar amounts of reactants at 20 °C. The reaction is shown in Figure 1. The formation of potassium glycidolyl was confirmed by 1H and 13C NMR (Bruker 300 MHz, Katowice, Poland) (Figure 2). Moreover, the macromolecules obtained were approved by the MALDI–ToF MS AXIMA Performance, Katowice, Poland) analysis (Figure 3). The following reaction was assumed. Theoretical and calculated mass from the mass spectra were summarized in Table 1. Glycidolyl molecule initiates the polymerization of potassium glycidoxide and joins a number of monomer units (propylene oxide, glycidol, monomer with carbazol groups), this time oxirane ring starting chain remains intact. In the next stage alkoxylate active center reacts with the ring, which leads to the formation of cyclic oligoether. At the same time, it creates a new alkoxylate center, which begins the chain growth.

The formation of proper polymeric macromolecules on the cyclic core was confirmed using MALDI ToF.

Branches are observed in case built into a subsequent inimer molecule in the chain following the reaction (Figure 1). The number of branches is equal to the number of potassium glycidoxide units and complexing agent to active centers ratio. One of the possible polymer branch structures formed in this way is presented in Figure 4. 

The molecular mas of selected star-shaped polymers was approved with MALDI-ToF analysis (Figure 5). The Mn of the polymer is equal to 9150 and the dispersity calculated by the use of SEC is equal to 1.45.

The structures of the utilized polymers are shown on the Figure 6, Figure 7, Figure 8 and Figure 9. Their characterization was the subject of the separate paper, “Synthesis and spectral analysis of hyperbranched poly(ethers) containing carbazole or hydrazone groups” [40]. For each sample geometric simulation was performed using Cambridgesoft ChemOffice software. For AP I (Figure 6), S XVI (Figure 7), SXVII (Figure 8), S XVIII (Figure 9) the simulation was performed for the structure with one side–arm to present clearly location of the side chain.

All measurements were performed on quartz microbalance type, Electrochemical Quartz Crystal Microbalance EQCM 5710 with flow-through holder 5610 (Figure 10). Analytical parameters of used QCMB are presented in Table 2.

The tests were carried out in a flow cell using a constant volume of solvent. The analyte in the form of a solution in an identical solvent was injected at points marked with red arrows, and changes in the natural frequency of the disk were observed in accordance with the Sauerbrey equation.
(1)$$∆f=\frac{-2f_{0\;}^{2}∆m}{A{\left(\mu_{Q}\rho_{Q}\right)}^{1/2}}$$
∆*f* = frequency change of a quartz resonator (MHz)∆*m* = mass change of a quartz resonator(g)*μ_Q_* = shear modulus of quartz (2.947 × 10^11^ dyn∙cm^−2^)*ρ_Q_* = quartz density (2.648 g∙cm^−3^)*A* = acoustically active area of a quartz resonator (πr^2^ = 3.14 × 0.25^2^ = 0.1963 cm^2^) *f*_0_ = resonant frequency of a quartz resonator (5 or 10 MHz for EQCM 5710)

The disk was placed so that the possible sedimentation of the analyte from the solvent would not distort the result by gravitational subsidence on the disk. The location of the disk guaranteed that only the absorbed analyte will be measured (its weight will be marked). 

Below is a schematic diagram of the cell as well as the measurement parameters, which presents the most important technical data of the device used.

The used quartz microbalance is a simple quartz resonator working with shear vibrations. Applying one or both transversely vibrating surfaces of the resonator to a certain loading mass reduces resonant frequency (extension of the vibration period) of the Kanazawa and decreases the amplitude of vibrations. In the case of analyzing the sensors, the amplitudes are very small—in the order of a fraction of nm, and the frequencies range from several to several dozen MHz. The frequency and amplitude of the microbalance can be measured electronically, but the frequency measurement is much more precise. The frequency change is proportional to the change in mass.

Quartz microbalances are used to measure the thickness of very thin layers of their mass and the detection of chemical compounds in liquids and gases. The microbalance acts as a detector of chemical compounds when the polyetherol layer, which is loading, is capable of selective sorption of particles from the environment. The drop in the vibration frequency of the microbalance is then proportional to the mass of the sorptionable particles.

The quartz microbalance enables particle detection at the ppm level depending on its center frequency and the molecular weight of the particles. The mass values determined with QCMB may be in the order of 0.1 ng = 10–10 g [41]. Sorption and particle desorption are in this case the law of Nernst division, which means that this type of sensor can work in a reversible manner.

The construction and principle of operation of the above-mentioned equipment is described in the manual, “Flow-through Quartz Crystal Holder type 5610 for Electrochemical Quartz Crystal Microbalance type EQCM 5710 User Manual”, prepared by Włodzimierz Kutner, Agnieszka Kochman [42].

## 3. Results and Discussion

Characterization of obtained polymeric materials was performed through a size exclusion chromatography technique. Number-average and weight-average molar masses, Mn and Mw respectively, and dispersity of polymers, Mw/Mn, were estimated by means of size exclusion chromatography (SEC) on a Shimadzu Prominance UFLC instrument at 40 °C on Shodex 300 mm × 8 mm OHpac column using tetrahydrofuran as a solvent. Polyethylene glycols (Fluka) were used as calibration standards. MALDI-TOF spectra were recorded on a Shimadzu AXIMA Performance instrument. Dithranol (Sigma-Aldrich) was used as a matrix without any cationating agent.

All the measurements for the adsorption analysis of novel designed and prepared polymeric stationary phase for drug analysis were prepared on QMCB. 

### 3.1. Polymer AP I

Test P001: medium was phosphate buffer pH 7.6; absorbed compound was acetyl salicylic acid dissolved in a buffer. The final concentration of acid was directly 50 mM (Figure 11).

Test P002: medium was phosphate buffer pH 7.6; absorbed compound was acetylsalicylic acid dissolved in ethanol. The final concentration of acid was about 50 mM. In order to verify the reaction of the polymer with ethanol, the last two instillations are only with ethanol. The study was conducted at elevated temperatures 40 °C (Figure 12).

Test P003: medium was phosphate buffer pH 7.6 with ethanol; absorbed compound was ethylenediamine dissolved in ethanol. The final concentration of acid was about 50 mM. In order to verify the reaction of the amine, polymer is added in two portions to ethylenediamine concentration of approximately 50 nM. The study was conducted at elevated temperatures, 40 °C (Figure 13).

### 3.2. Polymer XVI

Test P001: medium was acetonitrile; absorbed compound was acetylsalicylic acid dissolved in acetonitrile. The final concentration of acid was about 200 mM (Figure 14).

Test P002: medium was acetonitrile; absorbed compound was acetylsalicylic acid dissolved in acetonitrile. The final concentration of acid was about 150 mM. The measurement was performed to confirm the result of the first measurement (Figure 15).

### 3.3. Polymer XVII

Test P001: medium was THF; absorbed compound was acetylsalicylic acid dissolved in acetonitrile and aniline in THF. The final concentration of acid was about 265 mM (Figure 16) and final concentration of aniline was about 180 mM (Figure 17).

### 3.4. Polymer XVIII

Test P001: measurement was made on a flow microbalance; the medium was de-mineralized water; absorbed compound was ethylenediamine dissolved in water. Concentration of analyte step by step injected on the column in 5 stages refers 10 μM, 50 μM, 80 μM, 80 μM, and 80 μM, then the flow was increased at a rate of 1.5 mL/h to 25 mL/h. Frequency measurement was carried out together with the measurement of resistance (Figure 18).

Analyzing the obtained results, there is a quick and precise answer of the sensor to the presence of detected substance, whether these polymer complexes are sensitive to any of the analyzed small molar mass compounds. However, it was shown in previous papers that obtained polymeric sorbent in ambient conditions is not neutral. Unfortunately, with prepared only thin films of the polymeric phase, it is hard to obtain reproducible measurement results which prevent any misinterpretation. Small but visible lack of reproducibility was probably caused by differences in the thickness and uniformity of a polymer layer applied to the base surface. Thin films prepared by electrochemical deposition; spin-coating on organic, inorganic glass; and ITO will be the subject of a separate paper. The most representative behavior for each of the components is presented for API, XVI, XVII, and XVIII.

Sorbent API was very promising; unfortunately, without knowing the resistance, we cannot tell what caused sudden increase in frequency. XVI did not show any characteristic behavior for the complexing layer, but after dosing, decrease in frequency was observed (unfortunately the changes were random). XVII did not show the slightest ability to absorb acetylsalicylic acid or aniline. To be sure, we repeated the test with dynamic resistance measurement, and we did not see significant changes in resistance, which would indicate that nothing happened with the polymer layer. Despite of that, the frequency of quartz was still steadily increasing. Unexpectedly, sorbent XVIII gave the most promising results. With increased concentration of the absorbed compound, the frequency decreased. However, resistance increased, which gave us the information that the layer of polymer on the quartz is changing. Unfortunately, one measurement does not give us certain information about what is happening on the quartz surface.

## 4. Conclusions

Using the MALDI-ToF technique, the structure and molecular weight of the polymers obtained were confirmed; in addition, the behavior was consistent with the theoretical assumptions. In spite of the impossibility to apply the layer in an electrochemical manner, thin homogeneous layers were obtained, which did not go beyond the scope of the resonator operation, as can be seen in the figures shown. The applied layer was also sensitive to injections of small-molecule substances, which was demonstrated in all of the above graphs as changes in the center frequency of the vibrating disk. These results clearly indicate the sensitivity of the material to low molecular weight substances in a suitable solvent. Due to the stability of the layer, we used only polar solvents such as acetonitrile and buffer and were careful that the layer did not capture ions from the prepared buffers. In future experiments, it is recommended that the polymer layer be applied using spinning or centrifugal coating techniques, which can greatly unify the surface and allow for statistical analyses; it would also be necessary to deposit the layer from disks after measurement and re-perform the analysis using the time-of-flight detector to confirm the molecular weight change of the polymer by complexation of the corresponding compounds.

The polymers synthesized are perfect fitting to determine and complex small amines and low molar mass drugs, due to having some complexing properties that can be very useful in drug separation and identification, with high qualitative and quantitative sensitivity on molecular level. The number of the experiments and the lack of reproducibility of measurements cannot clearly define the size and performance capabilities of the supramolecular complexes; to determine this phenomena’s mechanism, more measurements are required with bigger repeatability. Measurements without resistance do not allow determining with certainty the absorption process during the experiment. Because mechanical properties of the polymer layer are a question during measurement that lasts at least one hour, it may be followed by swelling, stiffness, partial or complete washing out of layers from quartz. The main obstacle to clear and reliable analysis of the results was the lack of ability to conduction through the studied polymers. There was no possibility to apply the layer by electrochemical deposition. As a result, we have no information about the thickness, quality, and durability of the layer.

Assuming that the initial measurements were intended to refer to both the technique and the measuring equipment as well as to determine whether there are straight reactions between the layer of polymer and small molecular drug, we can qualify results as positive.

## Figures and Tables

**Figure 1 polymers-11-01554-f001:**
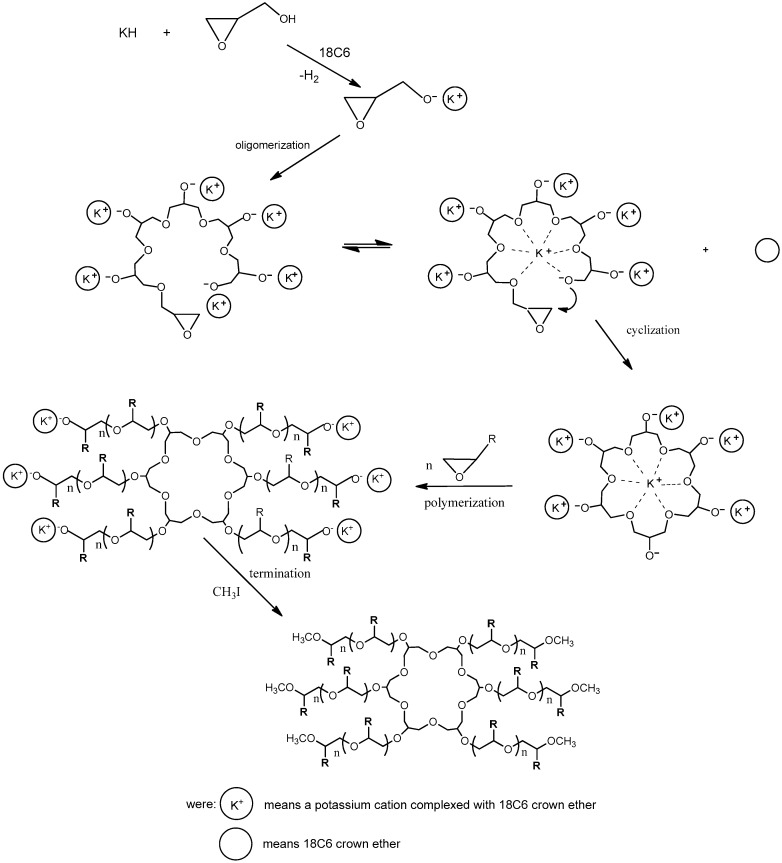
The reaction obtaining 6 arm star-shaped polymers.

**Figure 2 polymers-11-01554-f002:**
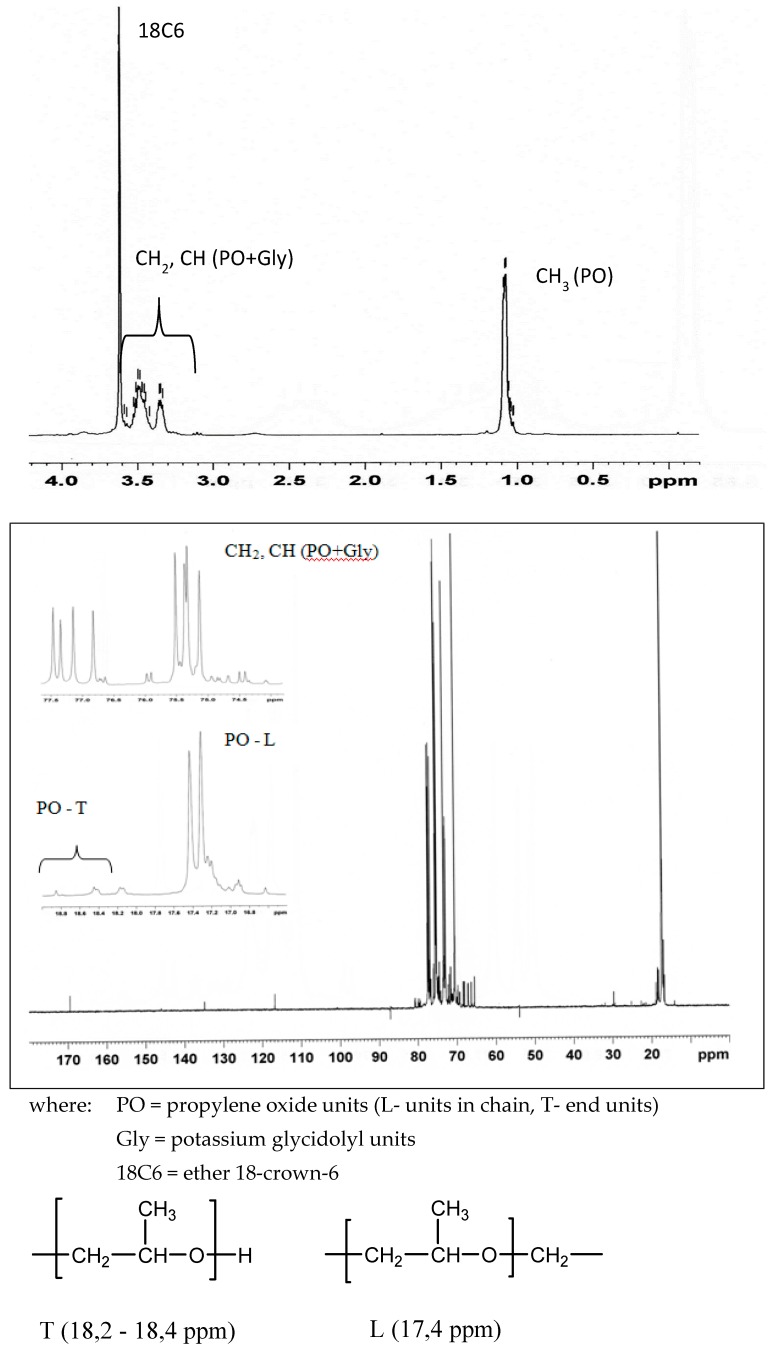
1H I 13C NMR spectra of obtained star-shaped macroinitiator with 6 arms in the presence of crown ether 18-c-6.

**Figure 3 polymers-11-01554-f003:**
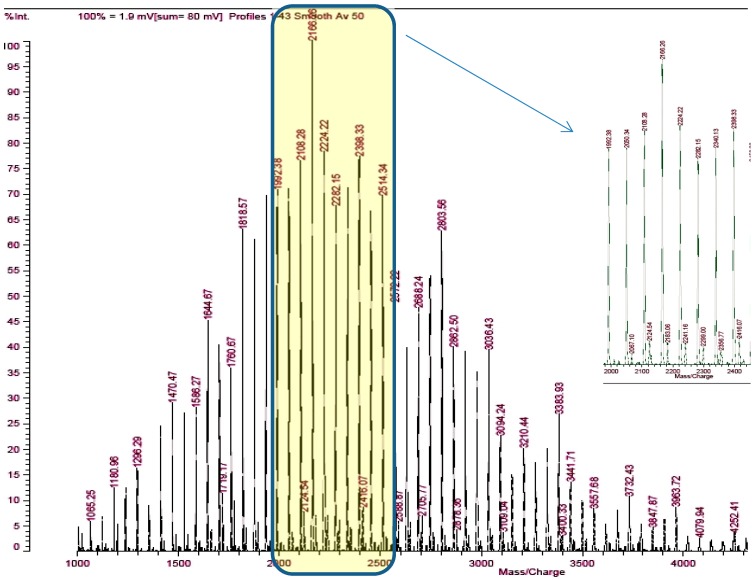
The MALDI-ToF analysis of obtained star-shaped structure on the cyclic macro-initiator with 6 alkoxide active centers.

**Figure 4 polymers-11-01554-f004:**
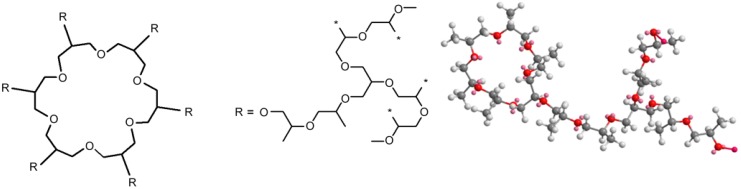
Simplified scheme macromolecule with two branches (* means places of chain growth).

**Figure 5 polymers-11-01554-f005:**
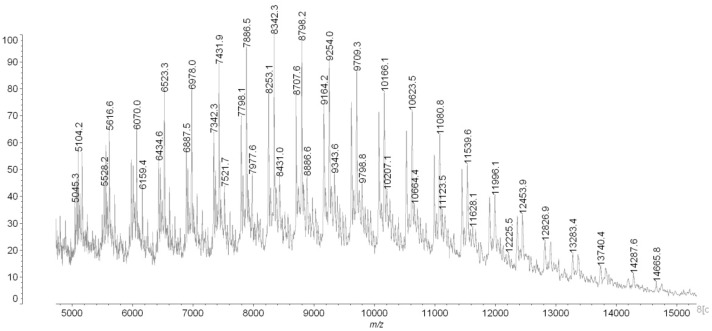
The MALDI-ToF mass analysis of AP1 Polymer.

**Figure 6 polymers-11-01554-f006:**
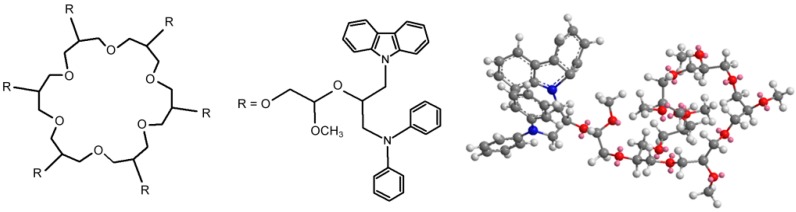
Structures of studied polymer—AP I Mn = 5300, D = 1.40.

**Figure 7 polymers-11-01554-f007:**
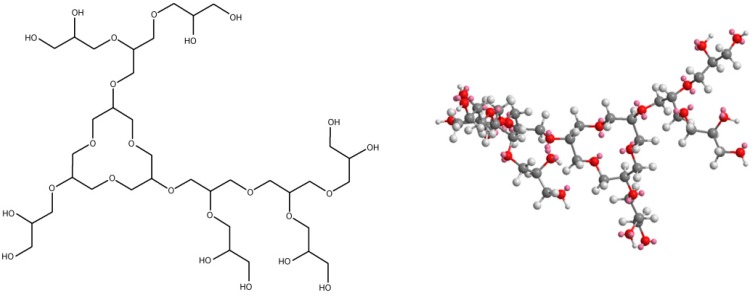
Structures of studied polymer—S XVI Mn = 3250, D = 1.21.

**Figure 8 polymers-11-01554-f008:**
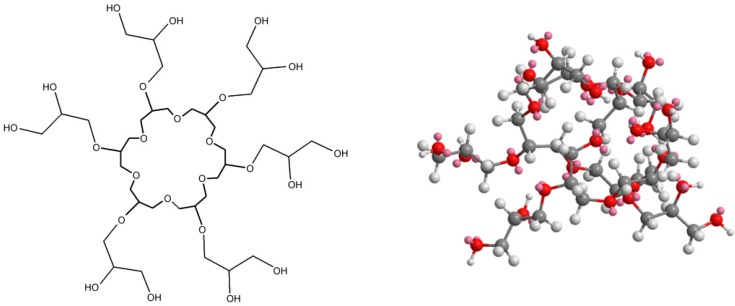
Structures of studied polymer—S XVII Mn = 6450, D = 1.45.

**Figure 9 polymers-11-01554-f009:**
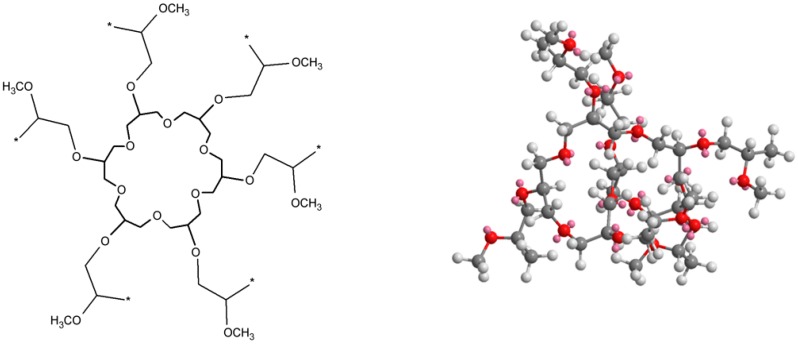
Structures of studied polymer—S XVIII Mn = 6850, D = 1.35 (* means places of chain growth).

**Figure 10 polymers-11-01554-f010:**
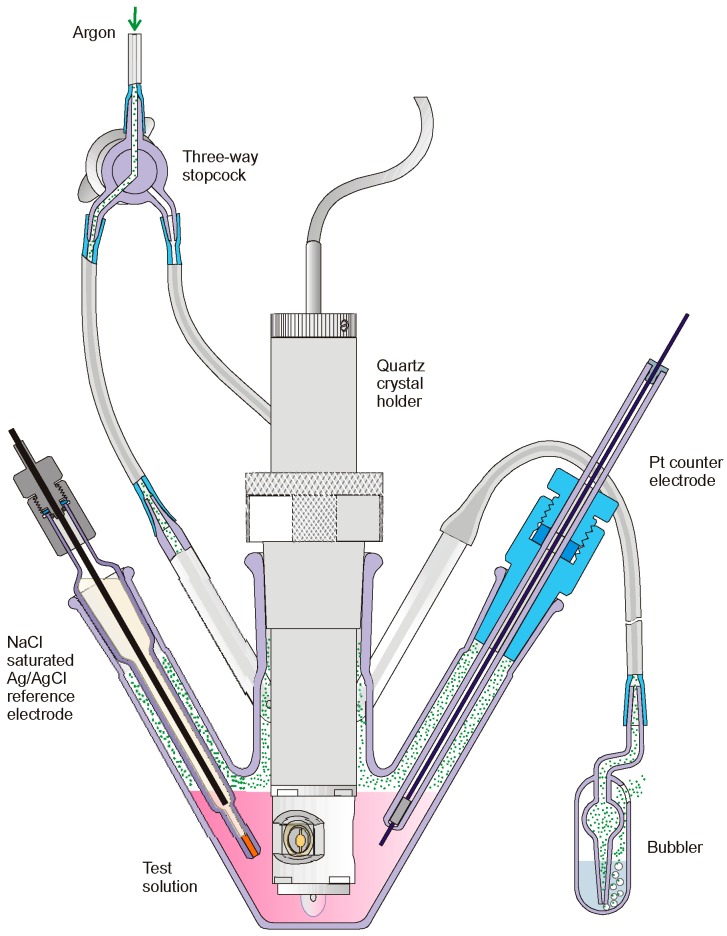
The analytical cell with holder equipped with quartz resonator.

**Figure 11 polymers-11-01554-f011:**
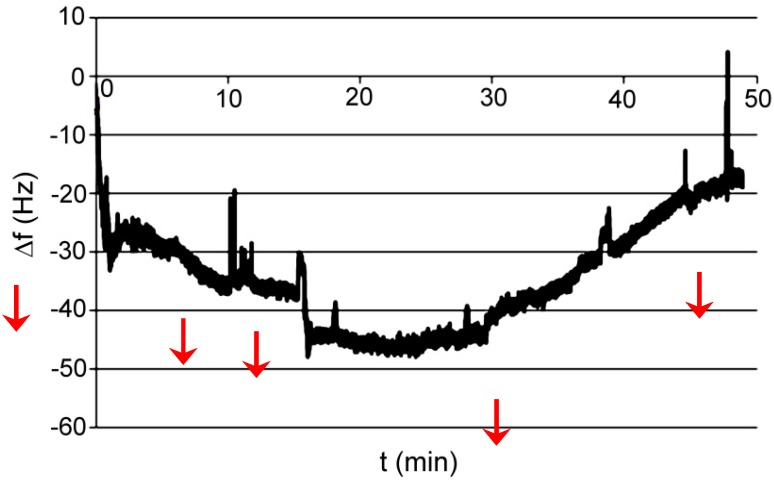
Changes of frequency in time for AP I polymer; absorbed compound—acetyl salicylic acid in a buffer.

**Figure 12 polymers-11-01554-f012:**
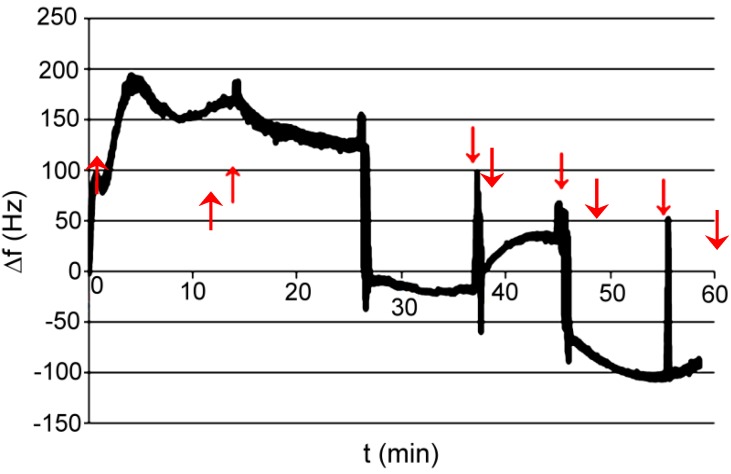
Changes of frequency in time for AP I polymer; absorbed compound—acetylsalicylic acid in ethanol.

**Figure 13 polymers-11-01554-f013:**
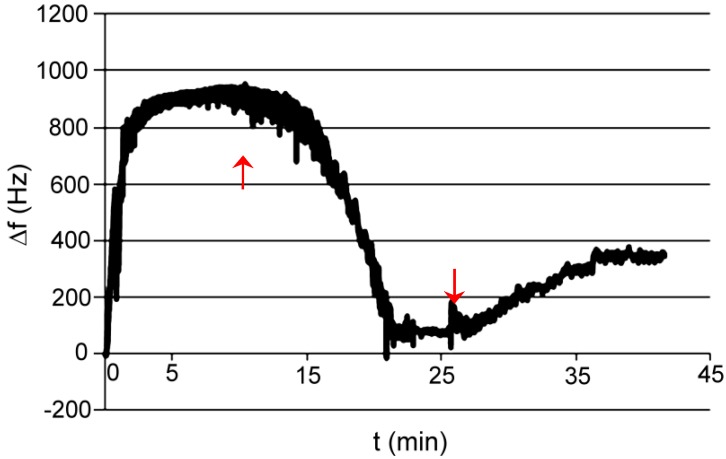
Changes of frequency in time for AP I polymer; absorbed compound—acetylsalicylic acid in ethanol.

**Figure 14 polymers-11-01554-f014:**
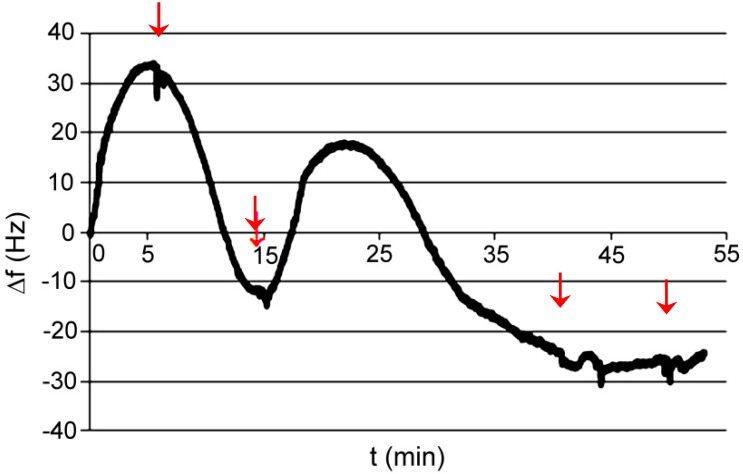
Changes of frequency in time for XVI polymer; absorbed compound—acetylsalicylic acid in acetonitrile.

**Figure 15 polymers-11-01554-f015:**
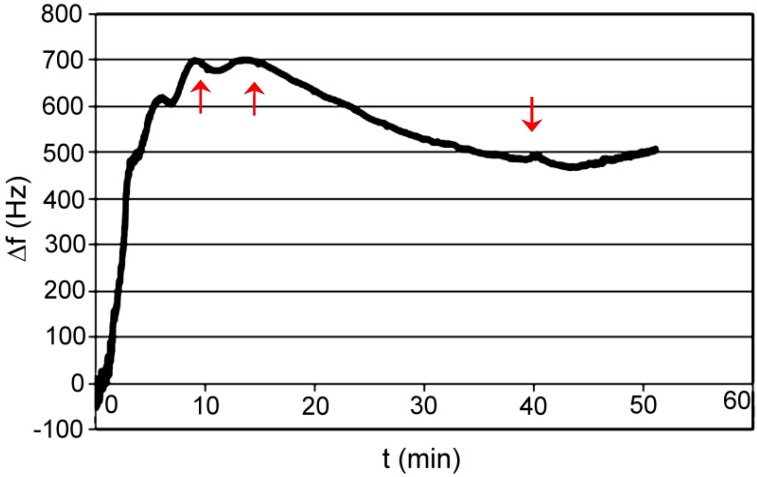
Changes of frequency in time for XVI polymer; absorbed compound—acetylsalicylic acid in acetonitrile.

**Figure 16 polymers-11-01554-f016:**
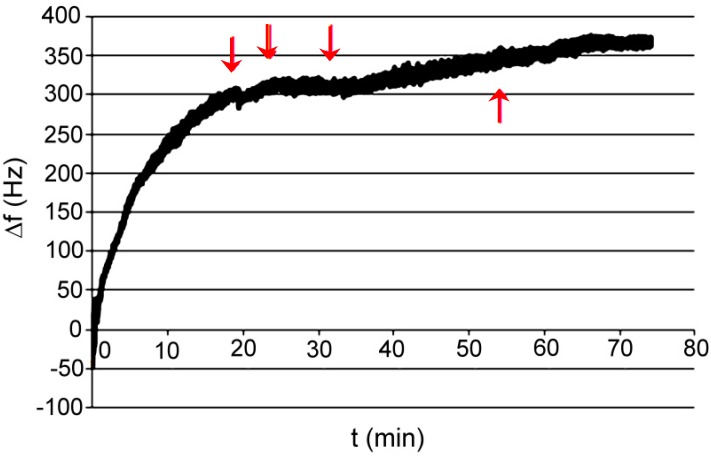
Changes of frequency in time for XVII polymer; absorbed compound—acetylsalicylic acid in acetonitrile.

**Figure 17 polymers-11-01554-f017:**
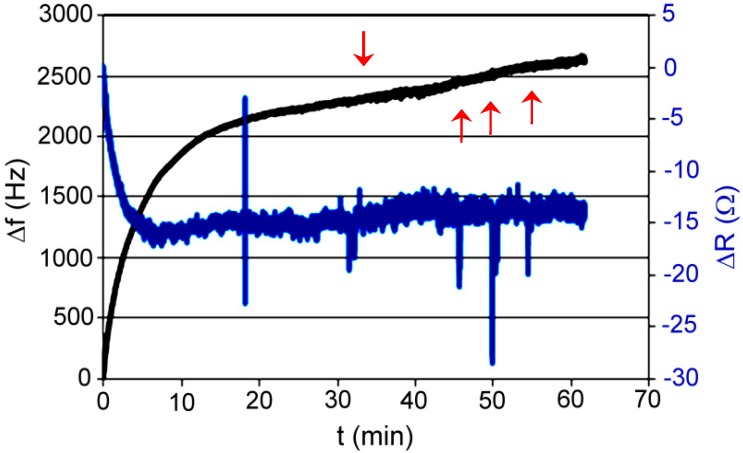
Changes of frequency in time for XVII polymer; absorbed compound—aniline in THF.

**Figure 18 polymers-11-01554-f018:**
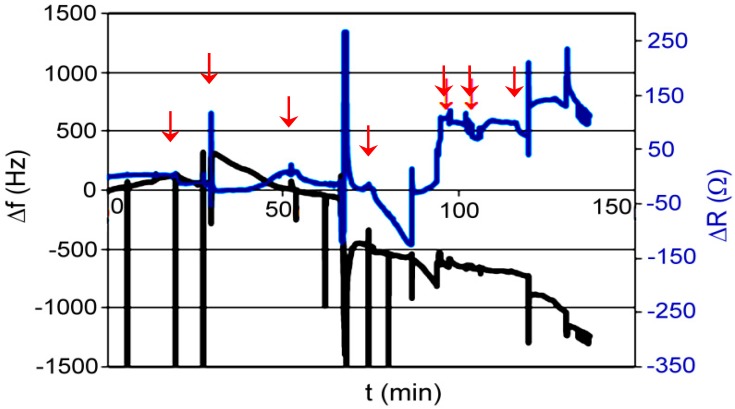
Changes of frequency in time for XVIII polymer; absorbed compound—ethylenediamine in water.

**Table 1 polymers-11-01554-t001:** The correlation between the theoretical and calculated masses from the MALDI-ToF spectra macromolecules.

Series	Potassium Adduct *	Number of Units	m/z Experimental	m/z Theoretical
A	G_x_(PO)_y_H_x_K^+^	x = 6, y = 27	2050.34	2050.42
		x = 6, y = 28	2108.28	2108.46
		x = 6, y = 29	2166.26	2166.5
		x = 6, y = 30	2224.22	2224.54
		x = 6, y = 31	2282.15	2282.58
B	G_x_(PO)_y_H_x_K^+^	x = 7, y = 26	2067.1	2066.42
		x = 7, y = 27	2124.54	2124.46
		x = 7, y = 28	2183.06	2182.5
		x = 7, y = 29	2241.16	2240.54
		x = 7, y = 30	2299	2298.58

* G = 1,2-dioxypropane units; PO = propylene oxide units.

**Table 2 polymers-11-01554-t002:** Quartz crystal parameters.

**Resonant Frequency, MHz**	5	10
**Surface Shape**	plano-convex	plano-plano
**Quartz Crystal Diameter, mm**	14.0	
**Electrode Diameter, mm**	5.0	
**Resonant Frequency Range, MHz**	4.90 to 5.05	9.90 to 10.05
**Sensitivity, ng/Hz cm^2^**	17.7	4.2
**Mass Resolution, ng**	0.35	0.08
**Detectability (at Signal/Noise = 3), ng**	3.3	1.1
**Maximum Mass Load, mg**	140	50
**Electrode Material**	Au, Ag, Pt, Pd or Ti; customized electrodes are also available (vacuum deposited or cathodically sputtered)	
**Quartz Crystal Holder**	Dip type; wetted parts are made of Kel-F^®^ and PTFE with the Au plated electrode contacts; diameter 25 mm; length 165 mm; different holders are necessary for 5 MHz and 10 MHz quartz resonators	

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
