# Peer review of "New Kind of Polymer Materials Based on Selected Complexing Star-Shaped Polyethers"

_polymers, 2019, doi:10.3390/polym11101554_

Round 1

Reviewer 1 Report

1.       In title as well as in Abstract and Keywords there still is word “polythioethers”, however, there is no data in the manuscript concerning polythioethers. Although Authors in response to the previous review write that the relevant data is included in the current version, I could not find them.

When someone agrees with comments in the review and writes that he has included respective data in the new version, he should do so.

2.       Abstract is still too long (only 200 words; see Instruction for Authors). In the Instruction is clearly stated that “The abstract should be an objective representation of the article: it must not contain results which are not presented and substantiated in the main text and should not exaggerate the main conclusions.”

When reading the manuscript one can see that the fragment “Potassium hydride was used for the first time as the initiator of oxiranes polymerization in 1995 for obtaining linear polyethers and polythioethers. Propylene oxide and selected oxirane monomers with carbazolyl in the substituent were selected as the monomers in this case and tetrahydrofuran as its solvent. The obtained polymers structures were characterized using the MALDI-TOF. It is found that in the initiation step potassium hydride deprotonates the monomer molecule and takes also part in the nucleophilic substitution. These reactions lead to the oxirane ring opening and formation of potassium allyloxide and potassium isopropoxide, respectively. Both alkoxides become real initiators of the polymerization and generate starting groups in the macromolecules. The polymerization is performed in the presence of 18-crown-6 in tetrahydrofurane (THF) reflux. In the initiation step only deprotonation of the monomer occurs. Moreover, nearly all unsaturation is represented by the cis- and trans-propenyloxy groups. The resulting polymeric material preferably cross-linked with selected di-oxiranes was then used as a stationary phase in the column and thin layer chromatography for amine separation and identification.” concerns rather the state-of-art and not results presented in manuscript.  
This problem was mentioned in my earlier review and Authors in the Answer to my remarks written that they corrected it.

3.       Authors writes about macrocyclic ethers but still uses the name “cryptands”. Macrocyclic ethers are usually cited as crown ethers. One should have in mind that complexation type is different with using crowns and cryptands. Also the structure of these compounds is different of course. (The expression “...cryptand like 18C6…” is not correct – see answer to review, it is crown ether 18C6!!!)

4.       Paragraph starting in line 72 as well as citations 38-48 concerns calixarenes not crown ethers

5.       Fig. 1 may be included into Fig 2; alone it is trivial – each reader knows how potassium hydride reacts with alcohols.

6.       The paragraph (line 108-113) is difficult to understand, maybe it is due to nomenclature of the KH - glycidol reaction products and it does not correspond to fig. 2 (by the way what does it mean the empty circle in the equilibrium reaction in first line of scheme???)

7.       Materials and Methods. There is no any materials as well as methods and protocols described in this chapter.
See the Instruction for Authors: “They should be described with sufficient detail to allow others to replicate and build on published results. New methods and protocols should be described in detail while well-established methods can be briefly described and appropriately cited.”
The description concerning the characterization of the materials used should be moved from Results and discussion to the Materials and methods.

8.       Authors writes that the polymerization was quenched by methylation however in the structures SXVI and S XVII there are no methoxy group present in structure. By the way, presented MALDI-ToF of “polymer” with carbazoyl groups which, as you write, are described earlier (W. Pisarski, A. Swinarew, B. Morejko, Z. Grobelny, A. 136 Stolarzewicz J.V Grazulevicius, V. Getautis, Synthesis and spectral analysis of hyperbranched poly(ethers) containing carbazole or hydrazone groups. Photonics Applications in Astronomy, Communications, Industry, and High-Energy Physics Experiments 2006; Why it is not cited in References???) but other structures are not characterized and should be described in this manuscript. In manuscript  Mn and MWD of investigated polymers have to be added but in Supplementary MALDI and NMR spectra as well as GC-MS experiment description should be presented.

Here I have more questions: (i) What is the driving force for formation of three glycidol units containing ring? (ii) Are the substances polymers or rather oligomers??? There is only one Mn (for AP1) and no data concerning other oligomers. (iii) what was the effect of molecules geometry simulation? Why they are relevant to this publication.

There is lack of comments concerning this problem.

9.       There is still unclear the connections between simple amines and aspirin absorption by presented material/polymers? and anti-doping procedures mentioned in the Abstract and Introduction. The use of ASA and especially amines as a model compounds in this study has to be explained.

The above comments, combined with those included in the review of the previous version of the work (in fact respective changes are not included in this version; reviewer 3), force me to recommend rejection of the manuscript.

Author Response

Comments and Suggestions for Authors

1.       In title as well as in Abstract and Keywords there still is word “polythioethers”, however, there is no data in the manuscript concerning polythioethers. Although Authors in response to the previous review write that the relevant data is included in the current version, I could not find them.

Answer: Following the reviewer suggestion we removed the term polythioeters. The term polythioeters found itself in the work because the authors work on a daily basis with polyethers and politioethers and it was entered because in the first phase of the article results on polytheoreters were to be presented, however, during the preparation of the work they were removed. 

2.       Abstract is still too long (only 200 words; see Instruction for Authors). In the Instruction is clearly stated that “The abstract should be an objective representation of the article: it must not contain results which are not presented and substantiated in the main text and should not exaggerate the main conclusions.”

Answer: As suggested by the reviewer, the abstract was reduced to 200 words and all additional results removed or moved to the results section. 

3.       Authors writes about macrocyclic ethers but still uses the name “cryptands”. Macrocyclic ethers are usually cited as crown ethers. One should have in mind that complexation type is different with using crowns and cryptands. Also the structure of these compounds is different of course. (The expression “...cryptand like 18C6…” is not correct – see answer to review, it is crown ether 18C6!!!)

4.       Paragraph starting in line 72 as well as citations 38-48 concerns calixarenes not crown ethers

Answer: According to the reviewer's suggestion, the nomenclature was changed from cryptand to crown ethers. All the sections concerning calixirane was removed. 

5.       Fig. 1 may be included into Fig 2; alone it is trivial – each reader knows how potassium hydride reacts with alcohols.

Answer: Figures 1 and 2 were combined as suggested by the reviewer 

6.       The paragraph (line 108-113) is difficult to understand, maybe it is due to nomenclature of the KH - glycidol reaction products and it does not correspond to fig. 2 (by the way what does it mean the empty circle in the equilibrium reaction in first line of scheme???)

Answer: For a better understanding of scheme 2 an empty circle was explained as a 18C6 crown ether molecule, where K + in a circle means a potassium cation complexed with 18C6 crown ether 

7.       Materials and Methods. There is no any materials as well as methods and protocols described in this chapter. See the Instruction for Authors: “They should be described with sufficient detail to allow others to replicate and build on published results. New methods and protocols should be described in detail while well-established methods can be briefly described and appropriately cited.” The description concerning the characterization of the materials used should be moved from Results and discussion to the Materials and methods.

Answer: According to the reviewer's suggestion, the microbalance description in the materials and methods section was completed and the description of materials was moved from the results section

8.       Authors writes that the polymerization was quenched by methylation however in the structures SXVI and S XVII there are no methoxy group present in structure. By the way, presented MALDI-ToF of “polymer” with carbazoyl groups which, as you write, are described earlier (W. Pisarski, A. Swinarew, B. Morejko, Z. Grobelny, A. 136 Stolarzewicz J.V Grazulevicius, V. Getautis, Synthesis and spectral analysis of hyperbranched poly(ethers) containing carbazole or hydrazone groups. Photonics Applications in Astronomy, Communications, Industry, and High-Energy Physics Experiments 2006; Why it is not cited in References???) but other structures are not characterized and should be described in this manuscript. In manuscript  Mn and MWD of investigated polymers have to be added but in Supplementary MALDI and NMR spectra as well as GC-MS experiment description should be presented.

Answer: According to the reviewers' suggestions, the work of W. Pisarski, A. Swinarew, B. Morejko, Z. Grobelny, A. Stolarzewicz J.V Grazulevicius, V. Getautis, Synthesis and spectral analysis of hyperbranched poly(ethers) containing carbazole or hydrazone groups. Photonics Applications in Astronomy, Communications, Industry, and High-Energy Physics Experiments 2006, was cited in references, and for other polymers, Mn and d were determined using gel permeation chromatography.

In response to a supplementary question, we state that glycidol spontaneously oligomerizes in a molar ratio of 1:1 with respect to crown ether, creating a cycle with 6 alcohol-active centers.

As suggested by the reviewer, information about the anti-doping action was removed from the text. 

9.       There is still unclear the connections between simple amines and aspirin absorption by presented material/polymers? and anti-doping procedures mentioned in the Abstract and Introduction. The use of ASA and especially amines as a model compounds in this study has to be explained.

Answer: According to the reviewer suggestions the appropriate section of the abstract were rebuild.

Reviewer 2 Report

1.                   The objective of this work and obtained results are not in complete agreement as the results are not clear at all.

2.                   What is the importance and motivation of this work is not clear?

3.                   Add details of all the chemicals (Source) and materials (Source) used in section 2.

4.                   There is no flow in the introduction, need revision.

5.                   Give more details of drug analysis by different polymer. In which state and how polymer and drug mixed?

6.                   I think the changes in frequency is not enough to estimate the drug sensing. Add more clear evidence for drug sensing.

7.                   Results and discussion is not clear need more revision. Explain all the figure with proper citation.

8.                   The Conclusion is not clear at all, make it simple and to the point.

Author Response

Comments and Suggestions for Authors

The objective of this work and obtained results are not in complete agreement as the results are not clear at all.

What is the importance and motivation of this work is not clear?

There is no flow in the introduction, need revision.

Answer: In the authors' opinion, in the abstract and in the introduction, there were indeed no smooth transitions between the research hypothesis and the presented results. This inaccuracy arose due to the fact that all authors work with star polymers and some reaction mechanisms as well as structures obtained in their course were obvious to the authors. The applied abbreviations could indeed cause the reader to misunderstand the topic. The abstract and introduction have been read and rebuilt so as to best introduce the reader into the work's scope and make it as understandable as possible and improve the correlation of the aim with the results.

Add details of all the chemicals (Source) and materials (Source) used in section 2.

Answer: In the second section, according to the reviewer's note, information on the reagents used was added.

Give more details of drug analysis by different polymer. In which state and how polymer and drug mixed?

I think the changes in frequency is not enough to estimate the drug sensing. Add more clear evidence for drug sensing.

Answer: The interaction between the polymers used and the low molecular weight substances used has been clarified. During the experiment, thin layers of star-shaped polymers were applied to the spindle disk by spining. The layers were prepared so that they do not overload the disk, i.e. not to change its natural frequency too much. Due to the star-shaped structure provided with appropriate arms, the complexation of the corresponding amines as well as other small molecule compounds was assumed through both mechanical trapping and chemical adhesion based on structural similarity. This phenomenon was confirmed with the use of the change in the natural vibration frequency of the microbalance, which confirm the deposition of additional mass on the vibrating disk.

13C NMR  spectra confirming the mechanism of the reaction were added.

The increase in mass due to the addition of low molecular weight compounds in an ultra-pure solvent could only be caused by adhesion of the low molecular weight material to the polymer layer. Frequency variations can also be caused by changes in the viscoelightness of the layer, however, because the low molecular weight compounds were added in the same medium in which the studies were conducted, the effect of the solvent which was given in microliter volumes on the viscoelasticity of the polymer layer was not suspected.

Results and discussion is not clear need more revision. Explain all the figure with proper citation.

The Conclusion is not clear at all, make it simple and to the point.

Answer: According to the reviewer's note, in order to better explain the results obtained, the measurements were discussed in detail in the results section and the interpretations are presented in the conclusions section.

Round 2

Reviewer 1 Report

Abstract:

First sentence: what does it mean “….ever-increasing importance of polymeric materials for low molecular weight compounds”?. What is the sense of this sentence?

….new sensors for selective small molecular mass complexation. Molecular mass cannot be complexed!!

It is found that in the initiation step potassium hydride deprotonates the monomer molecule and takes also part in the nucleophilic substitution. There is nothing about substitution reaction in the manuscript (also in Fig. 1 describing the polymerization process).

“The resulting polymeric material preferably cross-linked with selected di-oxiranes was then used as a ….”. There is no word concerning the cross-linking of these star-shape polyethers with dioxiranes!!!

Introduction:

In the aim of the work the selection of amines and ASA (amines and aromatic acid) should be shortly explain/justified.

Keywords:

Should be proposed new keywords replacing: amine desorption??; drug delivery system??; drug sensitive materials

Materials and Methods:

The chemical substances used in the synthesis should be more precisely characterized (purity, methods of purification before use). Form of KH (powder, suspension in mineral oil) should be presented. The reaction with KH may be carried out in THF but not in solution (KH is not dissolved in THF). Generally, this part is very laconic. Presented data in this place should contained sufficient detail allowing others to replicate and published results.

Once again, glycidolyl is substituent not a molecule. In the reaction of glycidol with KH potassium glycidoloxide/glycidoxide is formed and it is the initiator initiating the polymerization of other anions, i.e. glycidoxylates. By the way how the reaction of glycidol with KH proceeding in heterogeneous conditions is controlled??? What is the prove that there are no linear polyethers??? Moreover what is the driving force of the decomplexation of the potassium from its complex with 18C6??? Such not complexed potassium cation is necessary in further reaction where is the template for ring closure.

Results and Discussion

First paragraph, describing the analytical methods  should be moved into Materials and Methods. The information concerning the GC/MS analyses should be deleted because there no data concerning GC/MS analysis in manuscript. Moreover data concerning NMR analysis and of course spectrometer has to be added.

Authors write that in fig. 2 the NMR spectra of star-shaped macroinitiator is presented. Question is why there are signals derived from PO in macroinitaitor??? By the way the caption to fig. 2 is not correct; there are no typical information needed and presented for such spectra. Moreover all description should be in English.

The quality of Fig. 3 is unacceptable. It is almost impossible to analyze anything.

Table 1. Are the presented molecular masses obtained MALDI-ToF spectra somehow calculated or just read ???

Fig. 4: caption is: “Macromolecule with two branches” but looking one can see that there are six R substituents. Moreover above the Fig. Autors state that: “The number of branches is equal to the number of potassium glycidoxide units and complexing agent to active centers ratio.”. Does it means that there are only two glycidol molecules and what more?? were used in the preinitiation reaction?? Moreover Authors declare that in the preinitiation reaction reagents were used inn molar ratio as 1:1:1 glycilide to KH and 18C6. Please explain these doubts!!! (Whether a star-shaped two-arm polymer is not a linear polymer case!)

Line 184: the decimal point in English is a dot, not a comma.

Fig. 5. The spectrum should be discussed it means sequence lines should be ascribed to respective strictures.

Line 189: “The structures of the utilized polymers are shown on the Figs 6-9. Their characterization was the subject of the separate paper:…” Please give at least Mn and Đ in the caption of figures presenting the structure of used polymers!!!!!!

Lines 208-229: It is nice that Authors presents these data but in my opinion they should be placed in Supplementary (at least Table 2 and Fig. 10).

Lines 208-209 …”The analyte in the form of a solution in an identical solvent was injected at points marked with red arrows and….”. Where are these arrows???

The last paragraph od R&D (since line looks like conclusion. Is it possible to combine it with Conclusions?

Do not use: molecular mass or Mw/Mn (for molar mass distribution). Use molar mass and Đ!!!Also it is better to use 2-acetylsalicylic acid or acetylsalicylic acid than aspirin which is rather pharmaceutical name.

All work requires very thorough language verification by a polymer chemist - native speaker!!!

Author Response

Comments and Suggestions for Authors

Response to the Reviewer 1

Thank you for your constructive comments that helped us to increase the quality of our manuscript. We took all of your comments into consideration and indicated changes in the revised manuscript using red font.

Reviewer: Abstract – First sentence: what does it mean “….ever-increasing importance of polymeric materials for low molecular weight compounds”?. What is the sense of this sentence?

Answer: Polymeric materials are carriers or uptake of amines or drugs. We explained in the text.

Reviewer: ….new sensors for selective small molecular mass complexation. Molecular mass cannot be complexed!!

Answer: We agree with the expert reviewer and we have been corrected the sentence

Reviewer: It is found that in the initiation step potassium hydride deprotonates the monomer molecule and takes also part in the nucleophilic substitution. There is nothing about substitution reaction in the manuscript (also in Fig. 1 describing the polymerization process).

Answer: Oxiranes are usually polymerized via anionic polymerization. Potassium hydride (KH) is most frequently used catalyst for bulk i.e. propylene oxide anionic polymerization at 70–120 C yielding oligoetherpolyols for elastic polyurethanes. This process needs to use polyfunctional starter which usually contains 2–6 hydroxyl groups per molecule in this case mostly 6. The most important starters are ethylene glycol, dipropylene glycol and glycerol. Treating oxirane with propylene glycol in the presence of small amounts of KH oligoether diols are formed. Characteristic feature of oxiranes polymerization is the side reaction of hydrogen abstraction from the methyl group of monomers which results in the formation of allyl double bonds. In this reaction new chains are formed during the polymerization process. They take part in very fast cation exchange equilibrium reaction. All hydroxy chain-ends or alkoxy chain-ends propagate at similar rate. The fact is rather known therefore was omitted in the publication. In our opinion the reaction scheme including initiation process is well presented in the article.

Reviewer: “The resulting polymeric material preferably cross-linked with selected di-oxiranes was then used as a ….”. There is no word concerning the cross-linking of these star-shape polyethers with dioxiranes!!!

Answer: We agree with the expert reviewer and we have been corrected the sentence by adding the di-oxirane name and ratio

Reviewer: Introduction – In the aim of the work the selection of amines and ASA (amines and aromatic acid) should be shortly explain/justified.

Answer: We agree with the expert reviewer and the choice is shortly explain in the introduction.

Reviewer: Keywords – Should be proposed new keywords replacing: amine desorption??; drug delivery system??; drug sensitive materials

Answer: We agree with the expert reviewer and we added some new keywords.

Reviewer: Materials and Methods – The chemical substances used in the synthesis should be more precisely characterized (purity, methods of purification before use). Form of KH (powder, suspension in mineral oil) should be presented. The reaction with KH may be carried out in THF but not in solution (KH is not dissolved in THF). Generally, this part is very laconic. Presented data in this place should contained sufficient detail allowing others to replicate and published results.

Answer: We agree with the reviewer's opinion and made improvements in the form of a detailed description of the reagents used during polymerization, however, in response to the objection regarding the use of the word solution, it relates to the reaction mixture after the introduction of all reagents, then the complexation of potassium cation with the use of crown ether 18 Crown 6 a the complex is very well soluble tetrahydrofurane

Reviewer: Once again, glycidolyl is substituent not a molecule. In the reaction of glycidol with KH potassium glycidoloxide/glycidoxide is formed and it is the initiator initiating the polymerization of other anions, i.e. glycidoxylates. By the way how the reaction of glycidol with KH proceeding in heterogeneous conditions is controlled??? What is the prove that there are no linear polyethers??? Moreover, what is the driving force of the decomplexation of the potassium from its complex with 18C6??? Such not complexed potassium cation is necessary in further reaction where is the template for ring closure.

Answer: The reaction is carried out under homogeneous conditions due to the full solubility of the components after about 30 minutes spontaneous oligomerization occurs, which has already been published by the team and cyclization. Such a system has mainly 6 active alcoholate centers on which the chain propagates. No observation was made and nowhere mentioned that there is a release of potassium cation. The potassium cation remains complexed to the end and then removed in a column separation process.

Reviewer: Results and Discussion – First paragraph, describing the analytical methods should be moved into Materials and Methods. The information concerning the GC/MS analyses should be deleted because there no data concerning GC/MS analysis in manuscript. Moreover, data concerning NMR analysis and of course spectrometer has to be added.

Answer: We agree with the reviewer's suggestion and all the GC/MS information has been removed

Reviewer: Authors write that in fig. 2 the NMR spectra of star-shaped macroinitiator is presented. Question is why there are signals derived from PO in macroinitaitor??? By the way the caption to fig. 2 is not correct; there are no typical information needed and presented for such spectra. Moreover, all description should be in English.

Answer: The macroinitiator itself is stable only after glycidol oligomerization occurs. Then the forming metastable systems begin to disintegrate, therefore it is not possible to confirm the existence of the macro initiator itself having active alcoholic activated centers complexed with 18 Crown 6 ether. To confirm this what the reviewer required, it was necessary to complete the polymerization process but to connect at least one unit so that the system was stable. In this case, in order not to significantly affect the properties of the macrocycle itself and not hinder analysis, propylene oxide was chosen, hence the presence of propylene oxide units. Rest is corrected

Reviewer: The quality of Fig. 3 is unacceptable. It is almost impossible to analyze anything.

Answer: We agree with the expert reviewer but we don’t have a figure with a better quality so we decided to place the table 1 where we show the correlation between the theoretical and calculated from the MALDI-ToF spectra macromolecules masses.

Reviewer: Table 1. Are the presented molecular masses obtained MALDI-ToF spectra somehow calculated or just read???

Answer: Presented molecular masses obtained from MALDI-ToF are calculated.

Reviewer: Fig. 4: caption is: “Macromolecule with two branches” but looking one can see that there are six R substituents. Moreover, above the Fig. Autors state that: “The number of branches is equal to the number of potassium glycidoxide units and complexing agent to active centers ratio.”. Does it mean that there are only two glycidol molecules and what more?? were used in the preinitiation reaction?? Moreover, Authors declare that in the preinitiation reaction reagents were used inn molar ratio as 1:1:1 glycilide to KH and 18C6. Please explain these doubts!!! (Whether a star-shaped two-arm polymer is not a linear polymer case!)

Answer: The structure with only two branches was presented after filling with the monomer and chain propagation on only two of the six alcoholate active centers, which is clearly shown in the diagram. These types of abbreviated schemes are adopted when presenting chain propagation in case of structurally developed injectors so as not to blur the entire scheme.

Reviewer: Line 184: the decimal point in English is a dot, not a comma.

Answer: We agree with the expert reviewer and the mistake has been corrected.

Reviewer: Fig. 5. The spectrum should be discussed it means sequence lines should be ascribed to respective strictures.

Answer: The presented spectrum was presented only to show the molecular weight distribution of the polymer. Polymer materials presented in the work have already been researched, characterized and published. Structural analysis using nmdofof assisted nmr would take too much space at work, so in the authors' idea it was only important to present the molecular weight distribution and demonstrate that the tested material is in fact high-molecular.

Reviewer: Line 189: “The structures of the utilized polymers are shown on the Figs 6-9. Their characterization was the subject of the separate paper:…” Please give at least Mn and Đ in the caption of figures presenting the structure of used polymers!!!!!!

Answer: We agree with the expert reviewer and the information was completed.

Reviewer: Lines 208-229: It is nice that Authors presents these data but, in my opinion, they should be placed in Supplementary (at least Table 2 and Fig. 10).

Answer: The authors' intention was to present the difficult and little-known technique of mass measurement with the use of quartz microbalance. Therefore, in the authors' opinion, the scheme and table should remain in the main part of the article

Reviewer: Lines 208-209 …”The analyte in the form of a solution in an identical solvent was injected at points marked with red arrows and….”. Where are these arrows???

Answer: We agree with the expert reviewer and these arrows are placed in the proper figures.

Reviewer: The last paragraph od R&D (since line looks like conclusion. Is it possible to combine it with Conclusions?

Answer: We placed the last paragraph from R&D in Conclusions.

Reviewer: Do not use: molecular mass or Mw/Mn (for molar mass distribution). Use molar mass and Đ!!!Also it is better to use 2-acetylsalicylic acid or acetylsalicylic acid than aspirin which is rather pharmaceutical name.

Answer: We agree with the expert reviewer and we have been changed the names.

Reviewer: All work requires very thorough language verification by a polymer chemist - native speaker!!!

Answer: We agree with the expert reviewer and we verified the names. If the article will be accepted for publication, it will be verified by a native speaker - polymer chemist

Reviewer 2 Report

This article is still not suitable for publication in Polymers in its present form. It needs more revision and one more thing, author should be more careful during revision as all the comments were not answered.

1.                    What is the importance and motivation of this work is not clear?

2.                   There is no flow in the introduction, need revision.

3.                   Give more details of drug analysis by different polymer. In which state and how polymer and drug mixed?

4.                   I think the changes in frequency is not enough to estimate drug sensing. Add more clear evidence for drug sensing.

5.                   Results and discussion are not clear need more revision. Explain all the figure with proper citation.

6.                   The Conclusion is not clear at all, make it simple and to the point.

Author Response

Comments and Suggestions for Authors

Response to the Reviewer 2

Thank you for your constructive comments that helped us to increase the quality of our manuscript. We took all of your comments into consideration and indicated changes in the revised manuscript using red font.

Reviewer: What is the importance and motivation of this work is not clear?

Answer: We agree and we have been corrected the text adding the aim of the studies

Reviewer: There is no flow in the introduction, need revision.

Answer: The section was re-written

Reviewer: Give more details of drug analysis by different polymer. In which state and how polymer and drug mixed?

Answer: At the current stage of research, we do not yet describe the mechanisms associated with complexing - this is the beginning of a new work in which only the complexing mechanism will be presented. This mechanism is multi-stage and depends on ionic and thermodynamic factors. In addition, due to the lack of migration possibilities, this process is also diffusively controlled. Therefore, the multi-stage and complexity of this process does not allow us to accurately present it in this work. In the presented studies we only show that the described phenomenon is really urban and occurs in a predictable way for the analyzed chemicals

Reviewer: I think the changes in frequency is not enough to estimate drug sensing. Add more clear evidence for drug sensing.

Answer: To the best of our knowledge and after consultation with specialists from the Institute of Physical Chemistry, in particular the creator of the quartz microbalance on which we worked, professor cutter, this technique allows to clearly show whether there is an interaction between the substrate material located on a vibrating quartz disk and introduced into the reactor a low molecular drug research. There are many relationships linking a change in mass with a change in the disk's natural frequency if they are met, this means that additional material, additional mass, has been permanently associated with the ground. In our case, this is only possible by capturing low molecular weight substances in the flowing solution. In the opinion of the authors, this is sufficient evidence to confirm the described action of polymers. We don't know any other method that could confirm this thesis.

Reviewer: Results and discussion are not clear need more revision. Explain all the figure with proper citation.

Answer: The section was re-written

Reviewer: The Conclusion is not clear at all, make it simple and to the point.

Answer: The section was re-written

Round 3

Reviewer 1 Report

The article in actual version cannot be published.

I am not sure if I should prepare remarks for author because now, after three or four reviews many of my comments have not been included in the last version of the manuscript.

Once again:

Abstract is the summary of performed investigation. In the manuscript, the only information about the action of a hydride anion as a nucleophile is found in the abstract but in the reaction scheme this reaction is omitted. Similarly, information about the crosslinking of the polyethers tested was found only in the Abstract (the structures of the polymers used are shown and they lack any crosslinked fragments; third remark).

These problems were mentioned in last  review (remarks 3 and 4) and they are not solved.

My remark from last review:

Reviewer: Keywords – Should be proposed new keywords replacing: amine desorption??; drug delivery system??; drug sensitive materials

Answer: We agree with the expert reviewer and we added some new keywords.

R: There are no new keywords replacing these in version 2

Reviewer: Materials and Methods – The chemical substances used in the synthesis should be more precisely characterized (purity, methods of purification before use). Form of KH (powder, suspension in mineral oil) should be presented. The reaction with KH may be carried out in THF but not in solution (KH is not dissolved in THF). Generally, this part is very laconic. Presented data in this place should contained sufficient detail allowing others to replicate and published results.

Answer: We agree with the reviewer's opinion and made improvements in the form of a detailed description of the reagents used during polymerization, however, in response to the objection regarding the use of the word solution, it relates to the reaction mixture after the introduction of all reagents, then the complexation of potassium cation with the use of crown ether 18 Crown 6 a the complex is very well soluble tetrahydrofurane

R: There is no detailed description. Moreover I am sure (from my lab experience) that after adding the 18C6 into KH in THF mixture, KH is still insoluble or partially soluble.

Reviewer: Fig. 4: caption is: “Macromolecule with two branches” but looking one can see that there are six R substituents. Moreover, above the Fig. Autors state that: “The number of branches is equal to the number of potassium glycidoxide units and complexing agent to active centers ratio.”. Does it mean that there are only two glycidol molecules and what more?? were used in the preinitiation reaction?? Moreover, Authors declare that in the preinitiation reaction reagents were used inn molar ratio as 1:1:1 glycilide to KH and 18C6. Please explain these doubts!!! (Whether a star-shaped two-arm polymer is not a linear polymer case!)

Answer: The structure with only two branches was presented after filling with the monomer and chain propagation on only two of the six alcoholate active centers, which is clearly shown in the diagram. These types of abbreviated schemes are adopted when presenting chain propagation in case of structurally developed injectors so as not to blur the entire scheme.

R: If so, it is enough to present only one arm and to say that other are omitted.

Reviewer: Once again, glycidolyl is substituent not a molecule. In the reaction of glycidol with KH potassium glycidoloxide/glycidoxide is formed and it is the initiator initiating the polymerization of other anions, i.e. glycidoxylates. By the way how the reaction of glycidol with KH proceeding in heterogeneous conditions is controlled??? What is the prove that there are no linear polyethers??? Moreover, what is the driving force of the decomplexation of the potassium from its complex with 18C6??? Such not complexed potassium cation is necessary in further reaction where is the template for ring closure.

Answer: The reaction is carried out under homogeneous conditions due to the full solubility of the components after about 30 minutes spontaneous oligomerization occurs, which has already been published by the team and cyclization. Such a system has mainly 6 active alcoholate centers on which the chain propagates. No observation was made and nowhere mentioned that there is a release of potassium cation. The potassium cation remains complexed to the end and then removed in a column separation process.

R: My remarks contained in the first sentence repeated in every previous review were not taken into account. If I am wrong, please discuss this issue in response to the review. If I'm right, please include it in the manuscript. Regarding the last two sentences, look at the reaction scheme, first balance and formula after polymerization. Moreover reaction was quenched with methyl iodide.

Reviewer: Results and Discussion – First paragraph, describing the analytical methods should be moved into Materials and Methods. The information concerning the GC/MS analyses should be deleted because there no data concerning GC/MS analysis in manuscript. Moreover, data concerning NMR analysis and of course spectrometer has to be added.

Answer: We agree with the reviewer's suggestion and all the GC/MS information has been removed

R:GC/MS information is still present in R&D.

Ect. ect….

By the way in table 1: what is the origin of the polymer chain with x= 7 in B series? Macroinitiator generate six arms and initiation reaction is performed in mole ratio 1:1 KH/glycidol.

Moreover Authors suggests that the polyethers used in this investigations have been characterized in other paper, so it indicate that part concerning polymer used should be rewritten (If they were characterized they were synthesized and synthesis has been described). The chemistry of the synthesis as well as polymer characterization cannot be published second time. It is difficult to say which results have been published (I have no access to ref. 40).

The article needs to be completely rewritten to be published.

Author Response

Response to the Reviewer 1

Thank you for your constructive comments that helped us to increase the quality of our manuscript. We took all of your comments into consideration

Reviewer: Abstract is the summary of performed investigation. In the manuscript, the only information about the action of a hydride anion as a nucleophile is found in the abstract but in the reaction scheme this reaction is omitted. Similarly, information about the crosslinking of the polyethers tested was found only in the Abstract (the structures of the polymers used are shown and they lack any crosslinked fragments; third remark)

Answer: The abstract is written in a way that intentionally introduces the reader to information on applications and possible applications for starry Polyethers that are not typical polyetherols. Additionally, the work uses a very precise but little-known technique for measuring mass using the quartz microbalance developed at the Institute of Physical Chemistry in Warsaw. This type of microbalance is the only device that allows for a very precise analysis of the change in mass, these nuances make the reader fully understand the possibilities of new materials abstract in the opinion of the authors is and should be written to a large degree of generality, however, they fully introduce the reader to information about the material that is described later in this work.

Reviewer: There are no new keywords replacing these in version 2

Answer: We are sorry for that mistake, we replaced two keywords

Reviewer: from the previous remark

Materials and Methods – The chemical substances used in the synthesis should be more precisely characterized (purity, methods of purification before use). Form of KH (powder, suspension in mineral oil) should be presented. The reaction with KH may be carried out in THF but not in solution (KH is not dissolved in THF). Generally, this part is very laconic. Presented data in this place should contained sufficient detail allowing others to replicate and published results....

Answer: for the previous remark

We agree with the reviewer's opinion and made improvements in the form of a detailed description of the reagents used during polymerization, however, in response to the objection regarding the use of the word solution, it relates to the reaction mixture after the introduction of all reagents, then the complexation of potassium cation with the use of crown ether 18 Crown 6 a the complex is very well soluble tetrahydrofurane

Reviewer: There is no detailed description. Moreover I am sure (from my lab experience) that after adding the 18C6 into KH in THF mixture, KH is still insoluble or partially soluble.

Answer: In the authors' opinion, the reagents have been characterized in a necessary and sufficient way to repeat the synthesis because we do not know what we have not yet included in the description of the characteristics of the substrates used, we additionally include CAS number. Regarding the solubility of complex potassium cations with 18 Crown 6 to our best knowledge and conducted experiments, such a cation dissolves in everything including polar and non-polar solvents and the polymer itself as well as the monomer, these data have been confirmed many times during the synthesis because the reaction mixture before adding the monomer is homogeneous.

Reviewer: from the previous remark

Fig. 4: caption is: “Macromolecule with two branches” but looking one can see that there are six R substituents. Moreover, above the Fig. Autors state that: “The number of branches is equal to the number of potassium glycidoxide units and complexing agent to active centers ratio.”. Does it mean that there are only two glycidol molecules and what more?? were used in the preinitiation reaction?? Moreover, Authors declare that in the preinitiation reaction reagents were used inn molar ratio as 1:1:1 glycilide to KH and 18C6. Please explain these doubts!!! (Whether a star-shaped two-arm polymer is not a linear polymer case!)

Answer: for the previous remark

The structure with only two branches was presented after filling with the monomer and chain propagation on only two of the six alcoholate active centers, which is clearly shown in the diagram. These types of abbreviated schemes are adopted when presenting chain propagation in case of structurally developed injectors so as not to blur the entire scheme.

Reviewer: If so, it is enough to present only one arm and to say that other are omitted

Answer: The structure after changing is not clear enough for presentation. Regarding the presented structure, the presentation of the star structure of the polymer together with the side arms blur the structural image and structure and location of the central macrocycle. using only two arms introduces some misinformation which causes that the whole becomes incomprehensible and illegible after consultation within the team we have determined that according to the reviewer's suggestion we will add information under the diagram that the system is presented with only two chains due to improved readability.

Reviewer: from the previous remark

Once again, glycidolyl is substituent not a molecule. In the reaction of glycidol with KH potassium glycidoloxide/glycidoxide is formed and it is the initiator initiating the polymerization of other anions, i.e. glycidoxylates. By the way how the reaction of glycidol with KH proceeding in heterogeneous conditions is controlled??? What is the prove that there are no linear polyethers??? Moreover, what is the driving force of the decomplexation of the potassium from its complex with 18C6??? Such not complexed potassium cation is necessary in further reaction where is the template for ring closure.

Answer: for the previous remark

The reaction is carried out under homogeneous conditions due to the full solubility of the components after about 30 minutes spontaneous oligomerization occurs, which has already been published by the team and cyclization. Such a system has mainly 6 active alcoholate centers on which the chain propagates. No observation was made and nowhere mentioned that there is a release of potassium cation. The potassium cation remains complexed to the end and then removed in a column separation process.

Reviewer: My remarks contained in the first sentence repeated in every previous review were not taken into account. If I am wrong, please discuss this issue in response to the review. If I'm right, please include it in the manuscript. Regarding the last two sentences, look at the reaction scheme, first balance and formula after polymerization. Moreover reaction was quenched with methyl iodide.

Answer: The reaction occurs at a 1: 1 molar ratio of the complexing agent in the form of crown ether 18 crown 6 to alcohol active centers as previously described. The cycle is formed in the process of spontaneous oligomerization. The reaction, as noted by the reviewer, has been completed with methyl iodide so that the substitution process produces potassium iodide which is insoluble in the polymer or solvent which is tetrahydrofuran so that the resulting potassium iodide is sedimented and easy to remove from the reaction mixture without having to use processes silica gel column separation.

Reviewer: from the previous remark

Results and Discussion – First paragraph, describing the analytical methods should be moved into Materials and Methods. The information concerning the GC/MS analyses should be deleted because there no data concerning GC/MS analysis in manuscript. Moreover, data concerning NMR analysis and of course spectrometer has to be added.

GC/MS information is still present in R&D

Answer: for the previous remark

We agree with the reviewer's suggestion and all the GC/MS information has been removed

Reviewer: GC/MS information is still present in R&D

Answer: This time all the GC/MS information has been removed.

Reviewer: By the way in table 1: what is the origin of the polymer chain with x= 7 in B series? Macroinitiator generate six arms and initiation reaction is performed in mole ratio 1:1 KH/glycidol.

Answer: As shown above, the reaction with a molar ratio of 1 to 1 mainly produces six-arm cyclic structures, which means that a residual amount below the detection threshold for gel chromatography of 5- and 7-arm structures is also formed. This phenomenon is normal and related to the diffusive control of the initiation process.

Reviewer 2 Report

The manuscript has been carefully revised according to the reviewers' suggestion. The manuscript is worthy of publication in Polymers.

Author Response

We thank the expert reviewer

no further changes are required

Round 4

Reviewer 1 Report

My last remarks are marked as Reviewer LAST!!! (bolded)

Response to the Reviewer 1

Thank you for your constructive comments that helped us to increase the quality of our manuscript. We took all of your comments into consideration

Reviewer: Abstract is the summary of performed investigation. In the manuscript, the only information about the action of a hydride anion as a nucleophile is found in the abstract but in the reaction scheme this reaction is omitted. Similarly, information about the crosslinking of the polyethers tested was found only in the Abstract (the structures of the polymers used are shown and they lack any crosslinked fragments; third remark)

Answer: The abstract is written in a way that intentionally introduces the reader to information on applications and possible applications for starry Polyethers that are not typical polyetherols. Additionally, the work uses a very precise but little-known technique for measuring mass using the quartz microbalance developed at the Institute of Physical Chemistry in Warsaw. This type of microbalance is the only device that allows for a very precise analysis of the change in mass, these nuances make the reader fully understand the possibilities of new materials abstract in the opinion of the authors is and should be written to a large degree of generality, however, they fully introduce the reader to information about the material that is described later in this work.

Reviewer LAST: There is still no answer to: (1) In the manuscript, the only information about the action of a hydride anion as a nucleophile is found only in the abstract but in the reaction scheme this reaction is omitted. (2) Similarly, information about the crosslinking of the polyethers tested was found only in the Abstract (the structures of the polymers used are shown and they lack any crosslinked fragments; third remark). In abstract only information/results presented and discussed in manuscript may be placed!!

Reviewer: There are no new keywords replacing these in version 2

Answer: We are sorry for that mistake, we replaced two keywords

Reviewer LAST: Take into account also: quartz microbalance, adsorption

Reviewer: from the previous remark

Materials and Methods – The chemical substances used in the synthesis should be more precisely characterized (purity, methods of purification before use). Form of KH (powder, suspension in mineral oil) should be presented. The reaction with KH may be carried out in THF but not in solution (KH is not dissolved in THF). Generally, this part is very laconic. Presented data in this place should contained sufficient detail allowing others to replicate and published results....

Answer: for the previous remark

We agree with the reviewer's opinion and made improvements in the form of a detailed description of the reagents used during polymerization, however, in response to the objection regarding the use of the word solution, it relates to the reaction mixture after the introduction of all reagents, then the complexation of potassium cation with the use of crown ether 18 Crown 6 a the complex is very well soluble tetrahydrofurane

Reviewer: There is no detailed description. Moreover I am sure (from my lab experience) that after adding the 18C6 into KH in THF mixture, KH is still insoluble or partially soluble.

Answer: In the authors' opinion, the reagents have been characterized in a necessary and sufficient way to repeat the synthesis because we do not know what we have not yet included in the description of the characteristics of the substrates used, we additionally include CAS number. Regarding the solubility of complex potassium cations with 18 Crown 6 to our best knowledge and conducted experiments, such a cation dissolves in everything including polar and non-polar solvents and the polymer itself as well as the monomer, these data have been confirmed many times during the synthesis because the reaction mixture before adding the monomer is homogeneous.

Reviewer LAST: CAS are not needed. Simply write how the reagents were purified or write that all reagents were used as obtained from producer (KH was probably washed with THF or ….? and dried THF was dried ect.

Reviewer: from the previous remark

Fig. 4: caption is: “Macromolecule with two branches” but looking one can see that there are six R substituents. Moreover, above the Fig. Autors state that: “The number of branches is equal to the number of potassium glycidoxide units and complexing agent to active centers ratio.”. Does it mean that there are only two glycidol molecules and what more?? were used in the preinitiation reaction?? Moreover, Authors declare that in the preinitiation reaction reagents were used inn molar ratio as 1:1:1 glycilide to KH and 18C6. Please explain these doubts!!! (Whether a star-shaped two-arm polymer is not a linear polymer case!)

Answer: for the previous remark

The structure with only two branches was presented after filling with the monomer and chain propagation on only two of the six alcoholate active centers, which is clearly shown in the diagram. These types of abbreviated schemes are adopted when presenting chain propagation in case of structurally developed injectors so as not to blur the entire scheme.

Reviewer: If so, it is enough to present only one arm and to say that other are omitted

Answer: The structure after changing is not clear enough for presentation. Regarding the presented structure, the presentation of the star structure of the polymer together with the side arms blur the structural image and structure and location of the central macrocycle. using only two arms introduces some misinformation which causes that the whole becomes incomprehensible and illegible after consultation within the team we have determined that according to the reviewer's suggestion we will add information under the diagram that the system is presented with only two chains due to improved readability.

Rewieve LAST: There is still “Simplified scheme macromolecule with two branches” in Fig.4 caption. I hope “Simplified macromolecule scheme” will be enough but it is of course up to you.

Reviewer: from the previous remark

Once again, glycidolyl is substituent not a molecule. In the reaction of glycidol with KH potassium glycidoloxide/glycidoxide is formed and it is the initiator initiating the polymerization of other anions, i.e. glycidoxylates. By the way how the reaction of glycidol with KH proceeding in heterogeneous conditions is controlled??? What is the prove that there are no linear polyethers??? Moreover, what is the driving force of the decomplexation of the potassium from its complex with 18C6??? Such not complexed potassium cation is necessary in further reaction where is the template for ring closure.

Answer: for the previous remark

The reaction is carried out under homogeneous conditions due to the full solubility of the components after about 30 minutes spontaneous oligomerization occurs, which has already been published by the team and cyclization. Such a system has mainly 6 active alcoholate centers on which the chain propagates. No observation was made and nowhere mentioned that there is a release of potassium cation. The potassium cation remains complexed to the end and then removed in a column separation process.

Reviewer: My remarks contained in the first sentence repeated in every previous review were not taken into account. If I am wrong, please discuss this issue in response to the review. If I'm right, please include it in the manuscript. Regarding the last two sentences, look at the reaction scheme, first balance and formula after polymerization. Moreover reaction was quenched with methyl iodide.

Answer: The reaction occurs at a 1: 1 molar ratio of the complexing agent in the form of crown ether 18 crown 6 to alcohol active centers as previously described. The cycle is formed in the process of spontaneous oligomerization. The reaction, as noted by the reviewer, has been completed with methyl iodide so that the substitution process produces potassium iodide which is insoluble in the polymer or solvent which is tetrahydrofuran so that the resulting potassium iodide is sedimented and easy to remove from the reaction mixture without having to use processes silica gel column separation.

Reviewer LAST: The answer is not to my questions. There is still no answer to: Once again, glycidolyl is substituent not a molecule. In the reaction of glycidol with KH potassium glycidoloxide/glycidoxide is formed and it is the initiator initiating the polymerization of other anions, i.e. glycidoxylates as well as what is the driving force of the decomplexation of the potassium from its complex with 18C6??? Such not complexed potassium cation is necessary in further reaction where is the template for ring closure. Look at the reaction scheme, first balance and formula after polymerization.

Reviewer: from the previous remark

Results and Discussion – First paragraph, describing the analytical methods should be moved into Materials and Methods. The information concerning the GC/MS analyses should be deleted because there no data concerning GC/MS analysis in manuscript. Moreover, data concerning NMR analysis and of course spectrometer has to be added.

GC/MS information is still present in R&D

Answer: for the previous remark

We agree with the reviewer's suggestion and all the GC/MS information has been removed

Reviewer: GC/MS information is still present in R&D

Answer: This time all the GC/MS information has been removed.

Reviewer LAST:  OK. However, I have advised some time ago to move this first paragraph (part concerning methodology: Number-average and weight-average molar masses, Mn and Mw respectively, and dispersity of polymers, Mw/Mn, were estimated by means of size exclusion chromatography (SEC) on a Shimadzu Prominance UFLC instrument at 40 oC on Shodex 300 mm x 8mm OHpac column using tetrahydrofuran as a solvent. Polyethylene glycols (Fluka) were used as calibration standards. MALDI-TOF spectra were recorded on a Shimadzu AXIMA Performance instrument. Dithranol (Sigma-Aldrich) was used as a matrix without any cationating agent) to Materials and Methods.

Moreover, data concerning NMR analysis and of course spectrometer has to be added.

Reviewer: By the way in table 1: what is the origin of the polymer chain with x= 7 in B series? Macroinitiator generate six arms and initiation reaction is performed in mole ratio 1:1 KH/glycidol.

Answer: As shown above, the reaction with a molar ratio of 1 to 1 mainly produces six-arm cyclic structures, which means that a residual amount below the detection threshold for gel chromatography of 5- and 7-arm structures is also formed. This phenomenon is normal and related to the diffusive control of the initiation process.

Rewiever LAST: This problem is no mentioned in the manuscript when polymer used structures are described!! It should be shortly mentioned (in footnote to Table???) because you declare the six or three arm core of your star polyether (there is still question: what is the driving force/template for 3-active center macroinitiator formation?).

Additional remarks:

Figures 1 and 4: Please give the meaning of R substituent; if it is methyl group, derived from PO, the “*” cannot mean “places of chain growth”!!!!. The geometric simulation for all polymers was performed. What was the reason of such calculations? In the manuscript there is no any comment concerning this problem. In all manuscript D should be replaced with Ð. What does it means the “I” in Fig. 2 description??

In current version, the manuscript cannot be published in my opinion.

This manuscript is a resubmission of an earlier submission. The following is a list of the peer review reports and author responses from that submission.

Round 1

Reviewer 1 Report

The authors are addressing an interesting topic. However, the authors are required to improve the manuscript extensively before publishing in this journal. Both the experiment design and presentation are very poor.    

The language is poor to understand the manuscript. Extensive improvement in English is required. 

Also, there are major mistakes throughout the manuscript. For example, line 213... 80 uM, 80 uM, 80 uM. I do not understand what that means.

None of the results are presented in a manner that indicates the sensor is working. Based on these results it is very difficult to appreciate the conclusion. How the change in frequency changes with concentration?

A major revision is needed to explain the result section. Each figure presented required to explain in great details with scientific references. Explain the figures in details. It will be helpful if the details of the figures can be presented in the Figure section.

The present study cannot be published. It lacks the depth of results as we as the explanation. 

Reviewer 2 Report

The manuscript has regularly been written and results are interesting but many point need to be checked and it need significant revision before its acceptance in Polymers.

1.          The objective of this work and obtained results are not in complete agreement as the results are not clear at all.

2.          Add details of all the chemicals and materials used in section 2.

3.          There is no flow in introduction, need revision.

4.          Give more details of drug analysis by different polymer. In which state and how polymer and drug mixed?

5.          In my opinion, the changes in frequency is not enough to estimate the drug sensing. Add more clear evidence for drug sensing.

6.          Author are advised to add statistical analysis in Fig.7-14 .

7.          How different polymer materials have prepared is not clear.

8.          What is QCMB, is it the abbreviation of quartz microbalance?

9.          Results and discussion is too short, expand this section.  

10.        Also, carefully revise the manuscript for typographical and linguistic error.

Reviewer 3 Report

The reviewed manuscript concerns the sorption ability of new (?) polymer materials based on the selected complexing star-shaped polyethers and polythioethers.

In my opinion, the article is not suitable for printing and should be rejected. Below are detailed, though not all, comments concerning the reviewed manuscript.

1.       Title is inadequate to the content of the manuscript; there are no data concerning polythioethers.

2.       The abstract is too long. In addition, the information contained in the abstract is not confirmed in the manuscript; no NMR and MALDI-Tof results were included in the paper. There are also no specific information about the preparation of polymer materials to be tested. Abstract looks like it's about a different publication.

3.       Keywords: polythioethers – see p.1 of review; amine deorption or amine absorption??? Diethylene amine and aniline are not the drugs.

4.       Introduction: first paragraphs: the citations to work of Lutringhaus and Pedersen should be added. By the way the cryptands were first synthesized by Cram (cryptands  and crowns are not the same for both construction and complexation).

5.       Materials and methods is the chapter which should present the compounds and synthetic procedures in form enabling to other chemist to repeat the synthesis. There is no such information!!! This chapter presents the results of the research, unfortunately not those concerning the synthesis and characteristics of the polymers studied (NMR, MALDI-Tof, GC-MS and SEC).

6.       Fig. 1: the reaction is not proton abstraction but acid-base reaction. Presented reaction is trivial and may be omitted. However for belter understanding this part of manuscript Scheme presenting the polymerization should be shown. Line 107: what is potassium gycidolyl?? If it is the result of reaction of glycidol with KH the name should be potassium glycidoxide. What is gycidolyl molecule? Why there are no side reactions? Is it sure?

7.       Fig. 2: presented molecule possesses six branches.

8.       Next, in line 118, the authors write that the full characteristics of polymers is the subject of a publication printed in 2006. In this situation, the entire section devoted to chemistry should be removed, although molar masses should be placed in M&M (to be sure that they are the polymers), regardless of whether the samples tested are identical to those described in Pisarski et al. In such situation the “New” in title of the manuscript is strange.

9.       Why acetonitrile is used in presented analyses?? Why Authors use in Fig 7 acetylsalicylic acid (ASA) and next aspirin? Aspirin and ASA is the same.

10.   Authors suggest that the manuscript may be important in the aspect of the possibilities of applications in sport for drug and nutrition control. The use of ASA and especially amines as a model compounds should be explained. ASA is not a drug placed in the list of prohibited doping substances and what with the pyridine and ethylendiamine???

11.   Results and Discussion is very short and it is difficult to find “the success”. It does not present the importance of performed microbalance measurements.